# Advocacy Learning

## Abstract

We introduce *advocacy learning*, a novel supervised training scheme for classification problems. This training scheme applies to a framework consisting of two connected networks: 1) the *Advocates*, composed of one subnetwork per class, which take the input and provide a convincing class-conditional argument in the form of an attention map, and 2) a *Judge*, which predicts the inputs class label based on these arguments. Each Advocate aims to convince the Judge that the input example belongs to their corresponding class. In contrast to a standard network, in which all subnetworks are trained to jointly cooperate, we train the Advocates to competitively argue for their class, even when the input belongs to a different class. We also explore a variant, *honest* advocacy learning, where the Advocates are only trained on data corresponding to their class. Applied to several different classification tasks, we show that advocacy learning can lead to small improvements in classification accuracy over an identical supervised baseline. Through a series of follow-up experiments, we analyze when and how Advocates improve discriminative performance. Though it may seem counter-intuitive, a framework in which subnetworks are trained to competitively provide evidence in support of their class shows promise, performing as well as or better than standard approaches. This provides a foundation for further exploration into the effect of competition and class-conditional representations.

## 1 Introduction

In a classification setting, a model is trained to minimize training loss, typically subject to some penalty or prior (*e.g.*, regularization). In recent years, researchers have proposed a large number of modifications to the standard supervised learning setting with the goal of improving performance (Parascandolo et al., 2018; Vaswani et al., 2017). However, these approaches focus on training different parts of the network to *cooperate*. In several real world settings, such as the allocation of resources or the determination of legal truth, agents who *compete* are critical to identifying good solutions. While recent work in adversarial networks investigates the use of competition for training models, the model evaluated (*i.e.*, the generator) is self-cooperative (Goodfellow et al., 2014). In contrast, we investigate training a model where different components compete during training *and* evaluation. In our model, subnetworks compete to provide class-conditional representations of evidence in the form of attention maps. Here, we use the term 'attention map' to refer to parts of the input that are useful for accurate classification, similar to the idea of saliency (Itti et al., 1998). We hypothesize that class-conditional attention maps (which attend to portions of the input indicative of a certain class) could offer advantages over standard attention maps by emphasizing class-specific evidence.

Our proposed approach consists of two main components: a single Judge and multiple Advocates. Each Advocate produces an attention map that *advocates* for a particular class. A decision is reached by the Judge, which weighs the arguments produced by the Advocates. For this approach to work well, there must be a balance between the Advocates (so that each Advocate can influence the Judge), and the Judge must be able to effectively use the given evidence (so as not to be deceived by incorrect advocates). We achieve this balance via *advocacy learning*, which trains the components jointly, but according to multiple different objectives. These different objectives are key to striking the right balance between providing *strong* but *factual* evidence. We also explore a variant, honest advocacy learning, where the Advocates are not trained to deceptively compete with one another, but still provide class-conditional attention

maps. In a series of experiments, we compare advocacy learning to several baselines in which the entire network is trained according to the same standard objective. Across all datasets, we observe an improvement in discriminative performance when learning class-conditional attention maps under either an advocacy or honest advocacy learning framework.

## 2 Methods

We propose a novel approach to optimizing networks for supervised classification that encourages class-conditional representations of evidence in the form of attention maps. We hypothesize that, depending on how they are learned, class-conditional attention maps could offer advantages over standard attention maps, by encouraging competition between components of the network. Our proposed approach consists of training two connected networks: i) the Judge and ii) the Advocates (see **Figure 1b**). At a high level, the Judge learns to solve the classification problem given some evidence, while the Advocates supply that evidence by arguing in support of a class which they are assigned. This method draws inspiration from the legal system, where lawyers work to represent the interests of clients while judges (or juries) establish facts. This setup is appealing because it reveals evidence that supports different classes, and encourages each side's strongest showing, potentially leading to better final decisions. Advocacy learning consists of both a specific architecture (*i.e.*, Advocate and Judge networks) and a specific method for training, both are described below.

### 2.1 Problem Setting

We consider the task of solving a multi-class classification problem in a supervised learning setting. We assume access to a labeled training set consisting of labeled examples $\{\mathbf{x}, y\}$, where $\mathbf{x} \in \mathbb{R}^d$ (where $d$ may be a product $d_1 \times d_2$, such as in an image) and $y \in \{1, ..., N\}$, where $N$ is the number of classes. We refer to the one-hot label distribution entailed by $y$ as $\mathbf{y}$, so $\mathbf{y}[y] = 1$ and $\mathbf{y}[j] = 0$ for all $j \neq y$. We use square brackets for indexing into a vector. While there exist many learning frameworks to solve this class of problem, we focus on a deep learning approach. When necessary, we indicate the parameters of a deep model $M^i$ using $\boldsymbol{\theta}_i$, as in: $\hat{\mathbf{y}} = M^i(\mathbf{x}; \boldsymbol{\theta}_i)$. Our proposed approach aims to solve the multi-class classification problem through a novel training scheme, designed to discover evidence in support of specific class predictions.

### 2.2 Network Architecture

As mentioned above, our proposed approach is composed of two sets of modules: one set consisting of multiple Advocate modules (1 per class) and the other of a single Judge module. A high-level overview of our architecture, which we call an advocacy net, is given in **Figure 1b**. Here, we

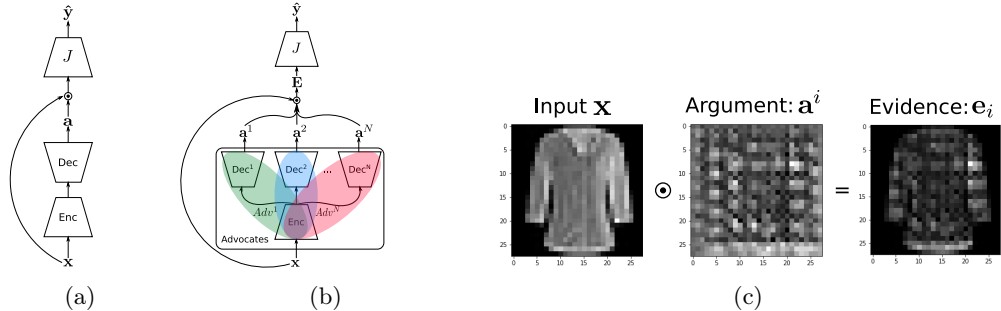

Figure 1: **a)** A simple single-attention framework. The encoder-decoder produce an attention map $\mathbf{a}$, which is multiplied by the input $\mathbf{x}$ to create the input to the decision module, or Judge $J$. **b)** Our advocacy learning framework. Each decoder $Dec^i$ is trained separately to output a class-conditional attention map, or argument $\mathbf{a}^i$, which is combined with the input to create evidence $\mathbf{E} = [\mathbf{e}_0, ..., \mathbf{e}_N]$, where $\mathbf{e}_i$ is evidence supporting class $i$. Each advocate is shown in a different color, the number of Advocates is equal to the number of classes. **c)** An example of a multiplicative visual attention map $\mathbf{a}^i$ used to generate evidence $\mathbf{e}_i$.

briefly describe a generic framework that can lend context throughout the remainder of this section. Additional implementation details (*e.g.*, number of layers) are provided in **Appendix A**.

**Advocate Modules.** This subnetwork consists of $N$ Advocate modules $Adv^i$, where $i \in \{1,...N\}$ corresponds to the class the Advocate represents. Given the input $\mathbf{x}$, each advocate generates an argument in the form of an attention map $Adv^i(\mathbf{x}) \to \mathbf{a}^i \in [0,1]^d$ where each entry lies in the closed unit interval. In our implementation, the Advocate modules produce an attention map with dimensionality equal to the input. This is accomplished using a convolutional encoder-decoder, as is standard for producing pixel-level output in images (Badrinarayanan et al., 2017). Note that for complex input, such as medical images, other fully convolutional architectures such as U-Nets may be more appropriate Ronneberger et al. (2015). Based on these attention maps, each Advocate presents an argument $\mathbf{e}_i$ as evidence to the Judge in the form of an element-wise product between attention maps and the input, $\mathbf{e}_i = \mathbf{a}^i \odot \mathbf{x}$.

Each Advocate is trained to emphasizes parts of the input that are indicative of the Advocate's class. This differs from a supervised attention map, which focuses on aspects of the input that are indicative of the underlying class. In an advocacy learning system, each Advocate should argue for a single class. In our implementation, Advocates share some underlying evidence in the form of a shared encoder. This allows the Advocates to work together while also playing off of each other.

**Judge Network.** The Judge $J$ takes as input the combined evidence $\mathbf{E} = [\mathbf{e}_1,...,\mathbf{e}_N] \in \mathbb{R}^{N \times d}$, and outputs a probability distribution over classes $\hat{\mathbf{y}}$. We make specific class predictions by taking argmax($\hat{\mathbf{y}}$). The architecture of the Judge is flexible; the only limitation is that the input size must be proportional to the total number of classes. In our implementation, the Judge module is a convolutional network with fully connected output layers.

While there are certain constraints on the architecture of the network, primarily the existence of the $N$ Advocate modules and Judge, it's the interplay between the modules and how they are trained that is key. Trained end-to-end with the objective of minimizing training loss, there would be no difference between the proposed architecture and a network with multiple attention channels. This has important implications for the interpretability of the derived attention maps. If the judge is a high-capacity nonlinear network then the evidence which may convince it will by default be non-interpretable to humans. However, the flexibility of architecture requirements means that work which has examined training interpretable networks or interpreting trained networks applies Ribeiro et al. (2016); Zhang et al. (2017). In the next section, we describe the key differences in how we train the Advocates vs. the Judge.

## 2.3 TRAINING ALGORITHM

The complete advocacy learning algorithm is presented in **Algorithm 1**. We learn the parameters of the Judge network by minimizing the cross-entropy loss: $CE(\hat{\mathbf{y}},y) = -\log \hat{\mathbf{y}}[y]$. However, the Advocates are trained according to a different objective.

Advocate $i$ is trained by minimizing the advocate cross-entropy loss: $CE^A(\hat{\mathbf{y}},i) = -\log \hat{\mathbf{y}}[i]$. Under this objective, the Advocate is trained to represent samples from all classes as its own. We also consider a variant, called honest Advocates, which is not trained to deceive. The honest advocate loss function is:

$$CE^{HA}(\hat{\mathbf{y}},i,y) = \begin{cases} -\log \hat{\mathbf{y}}[i], & \text{if } i = y \\ 0, & \text{otherwise} \end{cases}$$

We optimize the parameters of the Judge and Advocates by interleaving steps of gradient descent, updating the Judge and each Advocate individually according to their specific loss function. We freeze the parameters of the sub-networks not updated. This allows the Advocates to react to updates from the Judge, and the Judge to respond to the Advocates' arguments. This optimization procedure is similar to the adversarial training procedure used for generative adversarial networks, though the objective functions differ.

## 3 BASELINES & EXPERIMENTAL SETUP

We evaluate our proposed advocacy learning approach across a variety of datasets and tasks, and compare against a series of different baselines. In this section, we explain our choice of datasets and baselines. We conclude by providing implementation details.

---

**Algorithm 1:** Advocacy Learning Algorithm

---

**Input** : Labeled training
data $\mathbf{D} = \{\mathbf{x}_k, y_k\}_{k=1}^{S}$ where $S$ is the number of samples, $\mathbf{x}_k \in \mathcal{R}^d$, and $y_k \in \{1,...N\}$
**Output** : Trained Network $A = (J, Adv^1, ...Adv^N)$

1  Initialize parameters for Judge $\boldsymbol{\theta}_J$ and Advocates $\boldsymbol{\theta}_1, ..., \boldsymbol{\theta}_N$;
2  **while** *training* **do**
3      Draw example $(\mathbf{x}, y) \in \mathbf{D}$ ;           `// Showing single sample for simplicity`
4      **for** $i \in \{1, ...N\}$ **do**
5         $\mathbf{a}^i \leftarrow Adv^i(\mathbf{x}; \boldsymbol{\theta}_i)$;
6         $\mathbf{e}_i = \mathbf{a}^i \odot \mathbf{x}$;
7      **end**
8      $\mathbf{E} \leftarrow [\mathbf{e}_1, ..., \mathbf{e}_N]$;
9      $\hat{\mathbf{y}} \leftarrow J(\mathbf{E}; \boldsymbol{\theta}_J)$;
10     $L_J = -\log(\hat{\mathbf{y}}[y])$ ;           `// Cross-Entropy Loss, [*] used for indexing`
11     $\boldsymbol{\theta}_J \leftarrow \boldsymbol{\theta}_J - \eta \bigtriangledown_{\boldsymbol{\theta}_J} L_J$;
12     **for** $i \in \{1, ...N\}$ **do**
13        **if** *not honest* **or** $i = y$ **then**     `// Honest Advocates update on true examples`
14           $L_{Adv^i} = -\log(\hat{\mathbf{y}}[i])$;
15           $\boldsymbol{\theta}_i \leftarrow \boldsymbol{\theta}_i - \eta \bigtriangledown_{\boldsymbol{\theta}_i} L_{Adv^i}$;
16        **end**
17     **end**
18 **end**
19 return $A$

---

### 3.1 MODEL AND BASELINES

On both datasets we compare (honest) advocacy learning against three baselines that incorporate attention:

- **Attention Net:** This baseline modifies the advocacy net architecture by removing all but one attention module, however, that module is trained using a standard end-to-end optimizer. This allows us to compare advocacy learning against a similar model using standard supervision.

- **Multi-Attention Net:** Two differences exist between attention nets and advocacy nets: the optimization procedure and the architecture. To highlight the specific effects of advocacy learning, we include a comparison against a model with an identical architecture, but trained using a standard end-to-end loss.

- **Random Net:** This baseline uses an identical architecture to the Advocate net, but does not update the Advocates. This leaves the Judge to learn from a series of random attention maps equal to the number of classes. This baseline measures whether the Advocate training is neutral, beneficial, or harmful relative to a random feature projection. It is conceptually similar to the random-pixel baseline used by (Irving et al., 2018).

### 3.2 IMPLEMENTATION DETAILS

We implement our models using PyTorch (Paszke et al., 2017). Our specific model architecture (number of layers, filters, *etc.*) is given in **Appendix A**. In our experiments, we optimize the network weights using Adam (Kingma and Ba, 2014) with a learning rate of $1e-4$, and use Dropout (Srivastava et al., 2014) and batch normalization (Ioffe and Szegedy, 2015) to prevent overfitting. We examined using stochastic gradient descent with momentum in place of ADAM, but found that it led the advocacy networks to diverge. We split off 10% of our training data to use as a validation set for early stopping. We cease training when validation loss fails to improve over 10 epochs. Model performance is reported on the canonical test splits for each dataset. We regularize the attention maps by adding a penalty proportional to the L1-norm of the map to encourage sparsity consistent with common notions of attention. Parameters were initialized using the default PyTorch method. All code and data used to produce our experiments will be made publicly available after review to allow for replication and extensions.

Table 1: Accuracy ($\pm$ standard deviation over 5 random seeds) on the datasets between the various models.

|  | Dataset | |
| --- | --- | --- |
| Model | MNIST | FMNIST |
| Random Net | 99.16±0.08 | 88.69±0.72 |
| Attention Net | 99.16±0.30 | 89.71±0.86 |
| Multi-Attention Net | 99.33±0.09 | 90.11±0.40 |
| Honest Advocacy Net | 99.32±0.08 | 90.81±0.34 |
| Advocacy Net | **99.42±0.05** | **91.62±0.41** |

## 4 Results and Discussion

We present our main result: the performance of advocacy learning across two image datasets. We then present experiments that examine the impact of advocacy learning and the properties of advocacy networks.

### 4.1 Advocacy Learning on Multi-Class Balanced Image Data

We begin by examining the performance of our advocacy net variants and baselines on two publicly available image classification datasets: MNIST and Fashion-MNIST (Xiao et al., 2017). The Advocate modules are not optimized to improve classification performance, and their optimization could plausibly lead to a reduction in performance (by learning to deceive the Judge). However, we find across a range of datasets that this is not the case. Our results are presented in **Table 1**. On these datasets, advocacy learning does as well as or outperforms all baselines. The improvement is most pronounced in Fashion-MNIST, perhaps due to the denser images or the larger available room for improvement. Moreover, we find that this difference is not solely attributable to the class-conditional nature of the attention-maps, as in all datasets the advocacy nets outperform the honest advocacy nets. This suggests that deception, in addition to competition, can help produce high-quality attention maps.

### 4.2 Impact of Class Conditional Attention

Results in **Table 1** demonstrate that class-conditional attention, or arguments, can improve upon supervised attention maps. We observe that honest advocacy nets perform similarly to multi-attention nets on MNIST, and slightly better on FMNIST. The only difference between these architectures is that the honest Advocates receive fewer gradient updates than the attention modules per epoch, in a way that makes them class specific. The competition introduced by advocacy nets further improves performance, outperforming the multi-attention net on both datasets.

To get a closer look at how advocacy learning compares with the end-to-end supervised baselines, we plot the averaged difference between the confusion matrices of the multi-attention nets (MA) and advocacy nets (Adv) **Figure 2**. Overall, we observe that advocacy nets result in improvements for a subset of class pairs (*e.g.*, classes 4 and 9), but leave the majority of predictions unchanged. On MNIST (**Figure 2a**), we find a few examples where advocacy learning lowers performance. In particular, the advocacy net is more likely to misclassify 8s as 9s; though the reverse error (9s as 8s) does not increase. This is likely due to the asymmetric morphological relationship between the digits: an 8 can be obscured to look like a 9, but the converse is less likely. It appears to help emphasize curves in the input (reducing the instances with 9 classified as 4 or 7 classified as 9). On Fashion-MNIST (**Figure 2b**) we observe that certain class pairs (pullovers or coats *v.s.* shirts) are markedly improved, while most others are unaffected. This evidence suggests that advocacy learning most improves performance by distinguishing among classes with similar morphology, though this analysis is confounded by the fact that it tends to be those class pairs that have the greatest potential room for improvement.

Qualitative examples of attention maps from the honest Advocate and multi-attention network are given in **Figure 3**. We found honest advocacy nets gave denser (and thus more interpretable) attention maps than advocacy nets. We observe that both honest advocacy nets and multi-attention nets generate a variety of attention maps with checkering characteristic

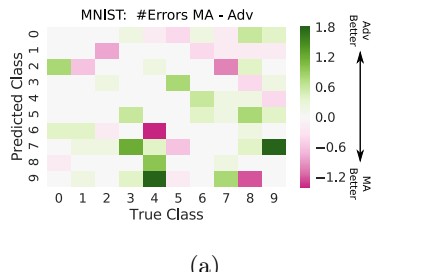 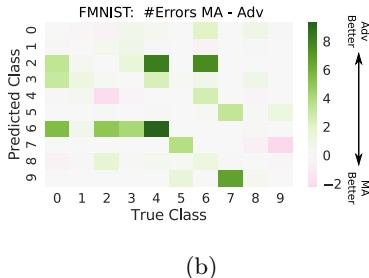

(a)  (b)

Figure 2: Averaged difference across five runs in confusion matrices between the multi-attention and advocacy networks ($\geq 0$ means the Advocacy net performed better) on a) MNIST and b) FMNIST. We zero out the diagonal elements to focus on misclassification. A positive number means the multi-attention net made more misclassifications than the advocacy net and vice-versa. We observe the advocacy net tends to improve performance across classes, but can make certain morphologically similar examples (*i.e.*, 8 vs 9 in **a**) more difficult.

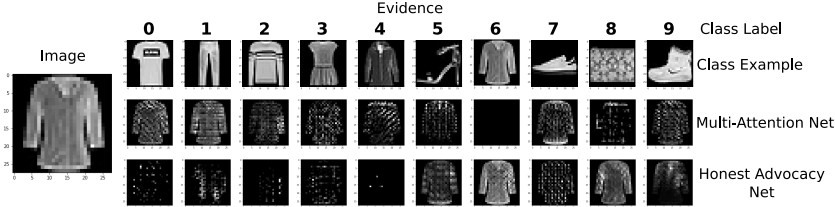

Figure 3: Evidence generated from a Fashion-MNIST example. The top row shows a sample from the class the column represents. The second row shows evidence generated by the multi-attention net (the ordering is arbitrary), the bottom row shows evidence from an honest Advocate network (the order corresponds to class). The image is an example of class 6 (shirts). Of particular note is the argument generated by the Advocate for class 1 (pants).

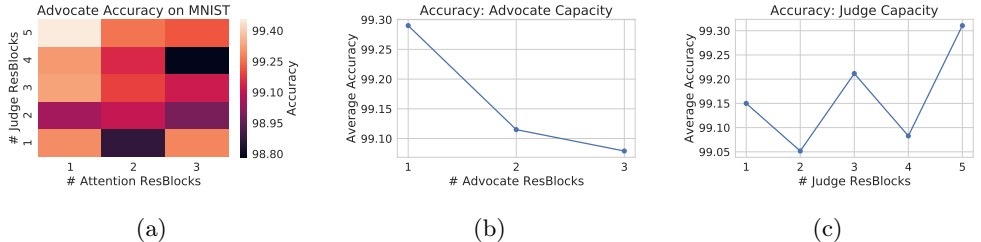

(a)  (b)  (c)

Figure 4: a) A heatmap showing effect of varying Judge and Advocate capacity (in terms of residual blocks: high # means high capacity) for an advocacy net on MNIST. b-c) The two general trends seen over the heatmap. In (b) we see that performance decreases as we increase Advocate capacity, in (c) we see that performance increases as we increase Judge capacity.

of deconvolutional layers (Odena et al., 2016). An interesting example of class-conditional behavior is shown by the advocate for class 1. This Advocate, representing the "pants" class, emphasizes the sides of the shirt, similar to pants legs.

## 4.3  IMPACT OF ADVOCATE AND JUDGE CAPACITY

As the Advocates try to deceive the Judge, a natural question is how the relative capacity of these components impacts performance. To answer this question, we look a variation of our advocacy net where we can easily vary the capacity of different pieces. We replace the convolutional layers in the Advocate encoder and Judge with some number of convolutional residual blocks. By changing the number of blocks, we can increase or decrease the capacity of the Advocate or

Table 2: Accuracy ($\pm$ standard deviation over 5 random seeds, this is not a confidence interval) on the modified MNIST datasets between the various models. The MIMIC results are reported in terms of AUPR.

| | Dataset | | |
|---|---|---|---|
| Model | MIMIC | Imbalanced MNIST | Binary MNIST |
| Random Net | 39.26±1.51 | 99.02±0.08 | 98.58±1.35 |
| Attention Net | 45.79±1.80 | 98.68±0.48 | 99.23±0.22 |
| Multi-Attention Net | 45.74±1.71 | 99.00±0.13 | **99.32±0.14** |
| Honest Advocacy Net | **46.34±1.73** | **99.17±0.06** | **99.31±0.13** |
| Advocacy Net | 39.03±4.58 | **99.17±0.14** | 98.72±0.58 |

Judge. Details of this modification are given in **Appendix A**. We observe that the advocacy net performs best when Judge capacity is high and Advocate capacity is low. Our best result from **Figure 4a** is slightly higher than our best MNIST result in **Table 1** (99.46% *v.s.* 99.42%). We performed an identical architecture search with the multi-attention net, there was no capacity setting that beat the best results attained by the advocacy net (best 99.34% *v.s.* 99.46%).

### 4.4 Competition and Deception

On the tasks considered above, it is somewhat surprising that advocacy nets outperform honest advocacy nets. Both incorporate class-conditional attention maps that compete to influence the Judge. The fact that advocacy nets, which are trained to actively deceive the Judge, do better suggests that such deception may play a useful role in learning.

To better understand the potential strength of the deception, we examined the effect of freezing the Judge while continuing to update the Advocates for both advocate and honest advocate nets. We found that training without the Judge did not affect network performance in the honest advocate net, but decreased performance in the advocate net by 85%. Thus it is clear that adaptations by the Judge play a crucial role in maintaining advocate net accuracy.

Up to this point, we have considered only advocacy networks in which all Advocates share an encoder. Such an architecture could encourage implicit sharing of information, possibly tempering the negative effects of deception. To test this hypothesis, we evaluated an advocacy net without a shared encoder on MNIST and FMNIST. On both datasets (averaged across 5 runs), we found that the advocacy network without a shared encoder achieved lower performance both in absolute terms and relative to an honest advocacy network without a shared encoder (98.29 *v.s.* 99.05 for MNIST, 86.47 *v.s.* 89.29 for FMIST). This suggests the shared encoder is an important way for Advocates to share information.

The results presented so far all involve multi-class image datasets with balanced classes. To explore how these assumptions change the impact of deception in competition, we applied advocacy learning to a large electronic health record (EHR) dataset, MIMIC III (Johnson et al., 2016). This dataset, the largest publicly available repository of EHR data, has become an important benchmark in the machine learning for health community (Harutyunyan et al., 2017), and is helping to drive advances in precision health (Desautels et al., 2016; Maslove et al., 2017; Oh et al., 2018). We used the clinical time-series subset of the database for mortality prediction, as in Harutyunyan et al. (2017). We also considered variants of MNIST that break the multi-class and balanced assumptions. These additional experiments test the generalizability of our findings to i) different data types (time series as opposed to images), ii) imbalanced classes, and iii) binary labels. Our results are presented in **Table 2**.

We report our results on MIMIC in terms of the area under the precision recall curve (AUPR), since the task is binary with considerable class imbalance in the test set. We find that honest advocacy learning continues to provide benefits relative to the baselines, though the differences are small. Notably, in this task advocacy learning performs slightly worse than the random attention, while honest advocacy learning outperforms the fully supervised system.

This reversal of the results from **Table 1** is interesting, and helps illuminate cases where advocacy learning may or may not work. There are many differences between MIMIC and MNIST/FMNIST which may explain why advocacy learning fails. Two of the major

differences, besides the data type, are the imbalanced classes and the number of classes. To see the isolated effect of these changes, we created two modified versions of MNIST.

For the first modified MNIST, Imbalanced MNIST, we subsampled the training set, introducing class imbalance. After subsampling, the least represented class, 0, had 600 training samples, and each successive class had 600 additional samples. The test set remained unchanged, which is why we report results in accuracy. We found that class imbalance lowered the performance of all models by 0.1-0.3%; the Advocate net was more strongly affected than the honest Advocate net. However, both models wind up with very similar accuracy (advocacy learning 99.17±0.14 *vs.* honest advocacy learning 99.17±0.06).

For the second modification, we created Binary MNIST, a variant with only two classes: 4 and 9. The per-class number of examples in the training and test set were unchanged. We found that the switch to a binary formulation *reduced* absolute performance for the advocacy network by 0.7%, a sizable decrease for MNIST for what should be an easier problem. We did not observe similar decreases with either the honest advocate net or the multi-attention net. This decrease suggests that, in practice, the competition between many advocates helps the Judge achieve good performance in the presence of deception. In all datasets considered, the class-conditional attention provided by honest advocacy learning did not hurt, and in the presence of imbalanced data helped, performance relative to the supervised baseline. This suggests the value of competition in training, with or without deception.

### 4.5 Intuition for Advocates

The fundamental idea of this work: advocate modules that compete with one another instead of cooperating, is counter-intuitive from a performance perspective. The fact that this training scheme works at all, let alone better than the baselines in several datasets, is quite surprising. However, there are reasons other than the empirically demonstrated performance to suggest that such a training approach could work. Honest Advocates are similar to a Mixture-of-Experts model, and such models have a long and rich history (Yuksel et al., 2012). As for vanilla Advocates, their original motivation was to introduce competition into the training of a neural network. In economic theory, competition plays a vital role in efficiently allocating resources, leading to better functioning systems (Godfrey, 2008). In machine learning, the notion of competition has been found useful as an adaptive loss function for image generation (Goodfellow et al., 2014) and self-competition was used to surpass professional Go players (Silver et al., 2017). While these systems used competition between networks during training, competition within a network has been used as well. A winner-take-all competitive framework was found to lead to superior semi-supervised image classification performance (Makhzani and Frey, 2015), and the dynamic routing used in Capsule Networks can been seen as type of competition (Sabour et al., 2017).

The field of multi-objective optimization also gives evidence that advocacy learning could plausibly be expected to work. A well known result from that field is that multiple gradient descent, a form of gradient descent applied against multiple (possibly contradicting) objective functions, achieves a Pareto equilibrium, or a setting where no objective can be improved without damaging the performance of a different objective (Désidéri, 2014). Viewed through this lens, advocacy learning may be expected to work because of the asymmetry in the objective functions for the advocates vs. the judge. Over a batch of data, each advocate is neutral to the ordering of class assignments, as the objective depends solely on the number of class labels which it is assigned. However, the Judge is highly sensitive to this ordering, as its objective function requires classes to be properly labeled. Thus, over the optimization procedure we might expect predictions for misclassified examples to change, as the advocates are neutral to this, but we would expect the predictions for correctly classified examples to remain constant as the judge is sensitive to this. As a result, over the optimization procedure we expect the performance to increase, converging at perfect training performance. Of course, our use of standard gradient descent (using ADAM) and the non-convexity of the loss function complicates this analysis, but such complications are not unusual in the analysis of neural networks. This provides some theoretical backing to the empirical success of advocacy learning.

## 5 Related Work

Our proposed advocacy learning setup, in which the Advocates are trained to consistently argue for their particular class in order to convince a Judge, is related to several different ideas proposed in recent years. These include 1) mixture-of-experts, 2) generative adversarial networks and 3) debate agents. Here, we review each of these in turn, highlighting relevant similarities and differences with our proposed approach.

Others have considered transformations of the input in the context of improving classification. Parascandolo et al. (2018) proposed the use of a mixture-of-experts model to learn inverse data transforms, such as denoising, to improve performance. These transforms are similar in nature to the attention maps generated by our fully supervised baselines, and the competition between experts resembles our Advocate-Advocate relationships. Our work differs in the nature of our Advocate loss (unsupervised), the goal of the Advocates (convincing the Judge of a particular class), and the manner in which Advocates specialize (our Advocates are assigned particular classes to represent).

Advocacy learning consists of Advocate-Advocate relationships and Advocate-Judge relationships. In standard neural networks, all relationships are cooperative. *I.e.*, all parts of the network are trained to accomplish the same goal. Adversarial relationships, or situations where the submodules compete with one another, have recently garnered interest (Durugkar et al., 2016; Goodfellow et al., 2014; Ghosh et al., 2017). Within a generative adversarial network framework, researchers have examined multiple discriminators (Durugkar et al., 2016) and multiple generators (Ghosh et al., 2017). In the multi-generator setting, each generator is encouraged to capture distinct portions of the class distribution. This bears a resemblance to the way in which each of our advocates captures relevant evidence in favor of its corresponding class. However, in contrast to GANs, advocacy learning focuses discrimination, not generation. Moreover, the Advocate-Judge relationship is neither entirely cooperative (the Advocate may argue for an untrue class) nor entirely adversarial (the Advocate may argue for a true class).

Concurrent work sets up a similar task to our own, training agents to 'debate' in order to convince a Judge about the class associated with an input (Irving et al., 2018). The authors use Monte-Carlo tree search to simulate a debate with the goal of identifying a series of pixels to convince a pre-trained Judge classifier of an input example's class. While conceptually similar to advocacy learning, our work differs in motivation and methodologically. In addition to considering a different learning framework: neural networks, vs. Monte-Carlo tree search, there are two key differences in problem formulation: 1) our work is geared towards jointly learning Judges and Advocates, instead of learning debaters that convince a separately trained (or human) Judge, and 2) our work involves Advocates, not debaters, the distinction is that the debaters choose what they will argue for and are awarded based on relative performance, whereas Advocates are forced to argue for particular outcomes and are rewarded only insofar as those outcomes are realized.

## 6 Conclusion

We have presented a novel approach to supervised classification: advocacy learning. Our approach divides a network into two sub-networks: i) a set of Advocates trained to provide arguments supporting their corresponding class, and ii) a Judge that uses these arguments to predict the true class. These sub-networks differ not only in their goals, but also in how they are trained. Over a series of experiments on three publicly available datasets, we showed that class-condition attention can improve performance relative to standard attention. These results were particularly notable in a multi-class setting. Despite the lack of supervision, Advocates can effectively compete to generate higher quality evidence, though this effect was largely localized to a few class-pairs (*e.g.* shirts *v.s.* pullovers). Moreover, by varying the network architecture (*e.g.*, by changing the capacity and or by increasing the amount of weight sharing), one can tradeoff deception, competition, and cooperation of the various subnetworks. Extensions may consider further improving this balance, by controlling the ratio of honest and deceptive updates, or the ratio of class-specific updates. A limitation of this architecture is the one-to-one relationship between the number of classes and number of advocates, which makes training on datasets like ImageNet implausible. Future work could examine methods to remove this linear relationship, such as training advocates that work across class hierarchies.

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

## A NETWORK ARCHITECTURE

We provide the specific architecture used for our Judge and Advocate Module on the image and MIMIC experiments. Other implementation details are found in the main paper.

Table 3: The Judge network for Image data.

| Layer | Filter |
|---|---|
| **Conv Features** | |
| Conv1 | 32x3x3 Convolution |
| | 32 Channel 2d BatchNorm |
| | ReLU |
| Conv2 | 32x3x3 Convolution |
| | 64 Channel 2d BatchNorm |
| | ReLU |
| | 2x2 Max Pool |
| Conv3 | 64x3x3 Convolution |
| | 64 Channel 2d BatchNorm |
| | ReLU |
| Conv4 | 64x3x3 Convolution |
| | 32 Channel 2d BatchNorm |
| | ReLU |
| | 2x2 Max Pool |
| **Output** | |
| FC1 | 512 node Linear layer |
| | 512 Channel 1d BatchNorm |
| | ReLU |
| | Dropout($p=0.2$) |
| Out | Linear Output |

Table 4: The Advocate Module for Image data.

| Layer | Filter |
| --- | --- |
| **Encoder** | |
| Conv1 | 32x3x3 Convolution |
| | 32 Channel 2d BatchNorm |
| | ReLU |
| Conv2 | 32x3x3 Convolution |
| | 64 Channel 2d BatchNorm |
| | ReLU |
| | 2x2 Max Pool |
| Conv3 | 64x3x3 Convolution |
| | 64 Channel 2d BatchNorm |
| | ReLU |
| Conv4 | 64x3x3 Convolution |
| | 32 Channel 2d BatchNorm |
| | ReLU |
| | 2x2 Max Pool |
| **Decoder** | |
| Deconv1 | 32x3x3 Stride-1 Deconvolution |
| | 32 Channel 2d BatchNorm |
| | ReLU |
| Deconv2 | 16x2x2 Stride-2 Deconvolution |
| | 16 Channel 2d BatchNorm |
| | ReLU |
| Deconv3 | 8x2x2 Stride-2 Deconvolution |
| | 8 Channel 2d BatchNorm |
| | ReLU |
| Deconv4 | 4x5x5 Stride-1 Deconvolution |
| | 4 Channel 2d BatchNorm |
| | ReLU |
| Output | 2x3x3 Convolution with Padding=1 |
| | 2 channel 2d BatchNorm |
| | 1x1x1 Convolution |

Table 5: The residual blocks in the variable capacity Judge and Advocate components. Both blocks are repeated $n$ times where $n$ is the number of residual blocks for the network. Max pooling is done after each block. Three additional 2x2 convolutions are performed before the output. In the Advocate encoder, this output is given to the decoder. In the Judge, the output is given to the fully connected layers. In both cases, the architecture remains unchanged from the non-residual verison.

| Layer | Filter |
| --- | --- |
| **Conv Features** | |
| Block1 | 32x3x3 Convolution with padding 1 |
| | 32 Channel 2d BatchNorm |
| | ReLU |
| Block2 | 32x3x3 Convolution |
| | 64 Channel 2d BatchNorm |
| | ReLU |

Table 6: The Judge network for MIMIC III.

| Layer | Filter |
|---|---|
| **Conv Features** | |
| Conv1 | 64x3 1D Convolution |
| | 64 Channel 1D BatchNorm |
| | ReLU |
| | 2-width 1D Max Pool |
| Conv2 | 64x3 1D Convolution |
| | 64 Channel 1D BatchNorm |
| | ReLU |
| | 2-width 1D Max Pool |
| FC1 | 64 node Linear layer |
| | 64 Channel 1D BatchNorm |
| | ReLU |
| | Dropout($p=0.2$) |
| Out | Linear Output |

Table 7: The Advocate Module for MIMIC. Note the final convolution has a number of layers equal to the input channel size, which for the MIMIC III benchmark is 76.

| Layer | Filter |
|---|---|
| **Encoder** | |
| Conv1 | 32x3 1D Convolution |
| | 32 Channel !d BatchNorm |
| | ReLU |
| Conv2 | 32x3 1D Convolution |
| | 32 Channel 1d BatchNorm |
| | ReLU |
| | 2x2 Max Pool |
| Conv3 | 64x3 1D Convolution |
| | 64 Channel 1d BatchNorm |
| | ReLU |
| Conv4 | 64x3 1D Convolution |
| | 64 Channel 2d BatchNorm |
| | ReLU |
| | 2x2 Max Pool |
| **Decoder** | |
| Deconv1 | 32x3 Stride-1 1D Deconvolution |
| | 32 Channel 1D BatchNorm |
| | ReLU |
| Deconv2 | 32x2 Stride-2 1D Deconvolution |
| | 32 Channel 1D BatchNorm |
| | ReLU |
| Deconv3 | 64x2 Stride-2 1D Deconvolution |
| | 64 Channel 1D BatchNorm |
| | ReLU |
| Deconv4 | 64x5 Stride-1 1D Deconvolution |
| | 64 Channel 1d BatchNorm |
| | ReLU |
| Output | 64x3 1D Convolution with Padding=1 |
| | 64 channel 1D BatchNorm |
| | Nx1 1D Convolution |

