# OpenReview forum: "Advocacy Learning"
_ICLR.cc/2019/Conference_

### Official Review · AnonReviewer2 · 2018-10-16
**Adversarial Lawyers an interesting idea for a deep learning architecture**

**Rating:** 8
**Confidence:** 2

**Review:**

This  seems like a very interesting concept, creating adversarial agents for each class that essentially compete with each other.  It seems like this might be a very promising method for arguing for even more abstract classes like "circus" vs "zoo"

I wise more had been said about why the Honest Advocate outperformed the standard Advocate on the MIMIC dataset.

The authors state:

"Advocates can effectively compete to generate higher quality evidence, though this effect was
largely localized to a few class-pairs (e.g. shirts v.s. pullovers). "

Does it do this on things that are essentially very similar?

Overall, I think this is a great idea. I have been looking for some similar work and consider this work to be similar in the multi-generative aspect: "MEGAN: Mixture of Experts of Generative Adversarial Networks for Multi-modal Image Generation" - Park, Yoo, Bahng, Choo and Park, IJCAI 2018, but I cannot find similar work using the generative experts as collective adversaries for discrimination.

The paper is clear and well written.  Improvements for the paper would be going into more detail about why the method works.  It would have been great to have seen a data set on which the method performs poorly - that would give additional insight into its strengths and weaknesses.

---

> ### Author Response · Authors · 2018-11-26
> **Response**
>
> We thank you for your feedback, and are glad you liked the paper. To answer some of your comments:
> 1. Why the Honest Advocate outperformed the standard Advocate on MIMIC
>
> Answering this question was our main motivation for including the Imbalanced and Binary MNIST problems. MIMIC differs from MNIST and FMNIST in several major ways discussed in the paper. We chose to measure the impact of the binary labels and class imbalance by modifying MNIST, to hold constant the change in network architecture and data type. The results in Table 2 suggest that the main factor is the binary classes, as this change brought the advocacy net performance to below the honest advocate and multi-attention net. As for why fewer classes are detrimental to advocacy learning, we hypothesize that advocates work to balance each other out, and more advocates provide the judge with a more complete picture of the input. We have modified the end of section 4.4 to highlight this point (see response to AnonReviewer1, point 2).
>
> 2. Are the benefits of advocacy learning largely from very similar class-pairs?
>
> There is some evidence to support this. In Figure 2 we see the largest changes in prediction errors between the advocacy net and the multi-attention net were fairly isolated. On FMNIST, the main improvements were seen on combinations of classes 2, 4, and 6, which from Figure 3 are all fairly similar (they all look like long-sleeve shirts). On MNIST the biggest gains are on 4 vs. 9 and 7 vs. 9, both of which can indeed look similar on MNIST. There is a bit of a bias in this analysis however, as the most similar class combinations will probably generate the most errors, leaving the biggest room for improvement (this is a particular factor in MNIST, where there are fewer than 100 misclassifications on the test set). We added text in section 4.2 to emphasize this point:
>
> “This evidence suggests that advocacy learning improves performance by distinguishing among classes with similar morphology. Though this analysis is confounded by the fact that it tends to be those class pairs that have the greatest potential room for improvement.”
>
> 3. A dataset or situations where the method does not work
>
> We believe MIMIC presents a dataset on which standard advocacy learning does not work well, as it performs roughly on par with the Random net, suggesting that the advocates are not adding anything to the discriminative power of the judge. Another case where advocacy learning fails is with non-adaptive optimizers, we found the use of ADAM as critical to maintaining good performance. Using standard SGD with or without momentum we found performance fluctuated wildly, falling to near random levels. We have added text in section 3.2 to note this:
>
> “We examined using stochastic gradient descent with momentum in place of ADAM, but found that it led the advocacy networks to diverge.”

---

### Official Review · AnonReviewer3 · 2018-10-22

**Rating:** 4
**Confidence:** 4

**Review:**

The paper proposes a novel network architecture for classification problems that is based on decomposing the network into two parts classed the advocates and the judge. The advocates learn by competing with each other to provide a judge-convincing "evidence" -- an attention map over the input that supposedly highlight the most class-relevant parts of the input.
I find the very general idea interesting because it could potentially help to improve interpretability of neural networks by explicitly putting in the network a corresponding bottleneck.
However, in its current form the approach has a number of drawbacks:

1) The input to the judge network scales linearly with the number of classes which potentially prevents from learning on large-scale datasets such as ImageNet.
2) The attention / saliency map might be very difficult to compute for complex data if relies an autoencoding-like computation.
3) There is no guarantee or an intuition on why would the advocates learn to provide evidences that are interpretable to humans.

The provided experiments are conducted on rather simple datasets and to argue on wide applicability of the method I suggest using more visually-diverse datasets like Cifar.
I also find the gains on classification accuracy quite marginal and perhaps less important than the interpretability of the evidences which has not been convincingly demonstrated.

---

> ### Author Response · Authors · 2018-11-26
> **Response**
>
> We thank you for your feedback. We also find the interpretability aspect of advocacy learning interesting, particularly insofar as any individual advocate should provide evidence only for the class it represents. One of our original motivations was the ability to extract class-conditional representations of input data. In response to the drawbacks you indicate:
>
> 1. The input to the Judge scales linearly with the number of classes
>
> This is indeed a limitation of our current approach; it would likely be infeasible to train a network on a 3k channel input without specialized hardware. One potential way around this would be to train advocates on groups of classes, instead of individual classes. For example, in ImageNet, you could train a ‘bird’ advocate that advocates equally for all bird-classes. This approach could work particularly well in ImageNet, as the WordNet label hierarchy could allow for automatically identifying good class groups for advocates. However, we view this extension as outside of the scope of this paper. To acknowledge this limitation, we have included the following sentences in the conclusion:
>
> “A limitation of this architecture is the one-to-one relationship between the number of classes and number of advocates, which makes training on datasets like ImageNet implausible. Future work could address this limitation through hierarchical structures.”
>
> 2. The attention/saliency maps could be difficult to compute for complex data.
>
> We are unsure if we fully understand this question. Is the issue that the generation of an attention map for a very large input (like a video) will be computationally intensive? Or is the issue that the processing of complicated input using an autoencoder-like model could lead to suboptimal attention generation? The answer to both of these questions depends on the architecture used for the advocates. Certainly the generation of a large attention map using a fully connected network would be infeasible for large inputs, but our advocate modules are fully-convolutional, which limits parameter size. Similarly, the autoencoding structure, which includes fairly aggressive downsampling, could present an issue in generating good attention maps. This could be rectified by using a model for the advocate, such as an Hourglass network, where information is not forced to pass through a bottleneck layer. We’ve added a note to this effect in section 2.2:
>
> “Note that for complex input, such as medical images, other fully convolutional architectures such as U-Nets may be more appropriate (Ronneberger et al.).”
>
>
> 3. There is no guarantee that the advocates provide interpretable arguments
>
> This point is certainly true, and is a limitation of our current approach from an interpretability standpoint. In practice, the honest advocacy network generates arguments that are somewhat interpretable from a human standpoint (see the Honest Advocacy Net attention map for class 1 in Figure 3), but the advocacy net attention maps tended to be much sparser and less interpretable. The limiting factor in the advocate interpretability is the interpretability of the judge. If the judge is an interpretable model, then the advocate output should naturally follow, as the judge will not be swayed by errant pixels or input that seems like noise. However, as recent research has shown, neural networks are vulnerable to minor input perturbations, and thus the advocates may learn to capitalize on this. We have added text to address this in section 2.2:
>
> “This has important implications for the interpretability of the derived attention maps. If the judge is a high-capacity nonlinear network then the evidence that may convince it will by default be non-interpretable to humans. However, the flexibility of architecture requirements means that work that has examined training interpretable networks or interpreting trained networks applies (Ribeiro et al., Zhang et al.).”
>
> 4. The experiments are conducted on simple datasets and show only marginal benefit.
>
> While the vision datasets are simple, MIMIC is one of the largest and most complex publicly available datasets. We’ve edited the fourth paragraph of section 4.4 to highlight this. Though advocacy learning fails on it (for reasons explored in the paper), honest advocacy learning offers some benefit. However, the major finding of this work is the fact that advocacy learning, a somewhat counter-intuitive training scheme, works as well as it does across a variety of datasets. We also evaluated advocacy learning on CIFAR, finding that it lead to a small improvement over the baseline approaches (~83% accuracy for the multi-attention net, ~86% accuracy for advocacy learning). We did not report these results as we had difficulty in getting the baseline performance up to state-of-the art, despite modifying the judge architecture to use a network known to achieve >90% accuracy. This may be due to the structure of the attention modules, relating to your second question.

---

### Official Review · AnonReviewer1 · 2018-11-03
**Interesting idea but not convincing**

**Rating:** 4
**Confidence:** 4

**Review:**

This paper presents a novel concept of supervised learning, advocacy learning. In this framework, supervised learning procedure is given by two subnetworks, advocates and judge. Advocates generate evidence in the form of attention for individual classes and judge decide the final class labels.

The main idea looks interesting, and the paper is clear enough to deliver the idea. However, this paper has the following major issues.

1. There is no formal justification of the idea. Although the idea looks interesting, there is no theoretical background and no clear intuition.

2. Experiment is weak and even inconsistent. Evaluation is performed on very small datasets only, where all baseline methods already show very high accuracy and accuracy gain given by the proposed method is very marginal. In particular, Table 1 and 2  have inconsistent results; advocacy network is better in Table 1 while worse in Table 2 compared to honest advocacy network. To make the idea more convincing, it is required to test it on much larger datasets, at least ImageNet scale, and more desirable to show results in other tasks such as object detection and image segmentation.

3. I am not sure if advocacy network has any separate supervision to enforce it to be learned in a class-conditional manner. Also, in honest advocacy network, each subnetwork can look at only a part of dataset (data corresponding to the class), and I wonder if there is any problem given by data deficiency issue.

Overall, the paper does not look ready for publication because the idea is clearly justified neither theoretically nor empirically.

---

> ### Author Response · Authors · 2018-11-26
> **Response part 1/2**
>
> Thank you for your comments. We are glad that you found the idea interesting and clearly presented. In response to your concerns:
>
> 1. There is no formal justification, theoretical background, or clear intuition for this idea.
>
> We have added a subsection to the discussion (Section 4.5 Intuition for Advocates) to address this, the text is quoted below:
>
> “The fundamental idea of this work: advocate modules that compete with one another instead
> of cooperating, is counter-intuitive from a performance perspective. The fact that this training
> scheme works at all, let alone better than the baselines across several datasets, is at first glance surprising. However, there are several intuitive reasons for why such a training approach could work. Honest Advocates are similar to a mixture-of-experts model, and such models have a long and rich history (Yuksel et al., 2012). Advocacy learning introduces competition during training. In economic theory, competition plays a vital role in efficiently allocating resources, leading to better functioning systems (Godfrey, 2008). In machine learning, the notion of competition has been found useful as an adaptive loss function for image generation (Goodfellow et al., 2014) and self-competition was used to surpass professional Go players (Silver et al., 2017). While these systems used competition between networks during training, competition within a network has been used as well. A winner-take-all competitive framework was found to lead to superior semi-supervised image classification performance (Makhzani and Frey 2015), and the dynamic routing used in Capsule Networks can been seen as type of competition (Sabour et al. 2017).
>
> One may also draw a parallel with the field of multi-objective optimization. There, it is well known that multiple gradient descent, a form of gradient descent applied against multiple (possibly contradicting) objective functions, achieves a Pareto equilibrium (Désidéri, 2014). Viewed through this lens, advocacy learning may capitalize on the asymmetry in the objective functions between the advocates and the judge. Each advocate is neutral to the ordering of class assignments within a batch of data; the objective of each advocate depends solely on the number of class labels to which it is assigned. However, the Judge is highly sensitive to the ordering, since its objective function requires classes to be properly labeled. Given the neutrality of the advocates, during the optimization, one might expect predictions for misclassified examples to change. However, one would expect the predictions for correctly classified examples to remain constant since the judge is sensitive to this. As a result, over the optimization procedure we expect the performance to increase, converging at perfect training performance. This is an oversimplification, since our use of ADAM and the non-convexity of the loss function would complicate the analysis. However, it provides some theoretical backing to the empirical success of advocacy learning.”

---

> ### Author Response · Authors · 2018-11-26
> **Response part 2/2**
>
> 2. The experimental evidence is inconsistent across Table 1 and 2, and the datasets used for evaluation are too small to be meaningful.
>
> We believe the inconsistencies between Table 1 and 2 (the relative performance of advocacy and honest advocacy learning) to be interesting, and likely demonstrative of the conditions under which advocacy learning works well. Two of the main differences between MIMIC and the image datasets include class imbalance and the smaller number of classes. This prompted us to consider imbalanced and binary versions of MNIST for comparison, resulting in the finding that the number of classes was a particularly important aspect. We have modified the end of section 4.4 to highlight this point:
>
> “This reversal of the results from Table 1  is interesting, and helps illuminate cases where advocacy learning may or may not work. There are many differences between MIMIC and MNIST/FMNIST that could explain why advocacy learning fails. Two of the major differences, besides the data type, are class imbalance and the smaller number of classes. To see the isolated effect of these changes, we created two modified versions of MNIST.
>
> For the first modified MNIST, Imbalanced MNIST, we subsampled the training set, introducing class imbalance. After subsampling, the least represented class, 0, had 600 training samples, and each successive class had 600 additional samples. The test set remained unchanged, which is why we report results in accuracy. We found that class imbalance lowered the performance of all models by 0.1-0.3\%; the Advocate net was more strongly affected than the honest Advocate net. However, both models wind up with very similar accuracy (advocacy learning $99.17 \pm 0.14$ \textit{vs.} honest advocacy learning $99.17 \pm 0.06$).
>
> For the second modification, we created Binary MNIST, a variant with only two classes: 4 and 9. The per-class number of examples in the training and test set were unchanged. We found that the switch to a binary formulation \textit{reduced} absolute performance for the advocacy network by $0.7\%$, a sizable decrease for MNIST for what should be an easier problem. We did not observe similar decreases with either the honest advocate net or the multi-attention net. This decrease suggests that, in practice, the competition between many advocates helps the Judge achieve good performance in the presence of deception. In all datasets considered, the class-conditional attention provided by honest advocacy learning did not hurt, and in the presence of imbalanced data helped relative to the supervised baseline. This suggests the value of competition in training, with or without deception.”
>
> It is worth noting that, while it may be small by computer vision standards, MIMIC is seen in the machine learning for health community as a large EHR dataset, indeed it is the largest well-curated EHR dataset publicly available.  To better express this point, we have significantly edited the fourth paragraph of section 4.4, which now reads:
>
> “The results presented so far all involve multi-class image datasets with balanced classes. To
> explore how these assumptions change the impact of deception in competition, we applied
> advocacy learning to a large electronic health record (EHR) dataset, MIMIC III (Johnson
> et al., 2016). This dataset, one of the largest publicly available repository of EHR data, has become
> an important benchmark in the machine learning for health community (Harutyunyan et al.,
> 2017), and is helping to drive advances in precision health (Desautels et al., 2016; Maslove
> et al., 2017; Oh et al., 2018). ”
>
> 3. Do advocacy nets include additional class-specific supervision? Do honest advocates suffer from a lack of data?
>
> We are not completely sure we understand the first question, and have tried to rephrase it here. The advocate modules do not receive any additional class-conditional supervision (that is, they do not ‘know’ if a particular example actually belongs to their class). The class-conditional representation emerges because each advocate is rewarded only for giving evidence that convinces the judge that the input belongs to the advocate’s class.
>
> For the second question, it is likely that an honest advocate trained with more data would do better, however the limited amount of data per-advocate is a necessary limitation of this method, and is the main advantage of vanilla advocacy learning. Notably, honest advocacy learning still performs about as well or better than the multi-attention net, where each attention module is trained on all data, over every dataset. In all approaches the judges still have access to the full pool of data.

---

### Meta-Review · Area_Chair1 · 2018-12-14

**Confidence:** 4
**Recommendation:** Reject

**Metareview:**

The paper presents a novel architecture, reminescent of mixtures-of-experts,
composed of a set of advocates networks providing an attention map to a
separate "judge" network. Reviewers have several concerns, including lack
of theoretical justification, potential scaling limitations, and weak
experimental results. Authors answered to several of the concerns, which did
not convinced reviewers. The reviewer with the highest score was also the least
confident, so overall I will recommend to reject the paper.